# Healthy dynamics of CD4 T cells may drive HIV resurgence in perinatally-infected infants on antiretroviral therapy

**Sinead E. Morris[1], Renate Strehlau[2], Stephanie Shiau[3], Elaine J. Abrams[4,5,6], Caroline T. Tiemessen[7], Louise Kuhn[4,8], Andrew J. Yates[1]\*, on behalf of the EPIICAL Consortium and the LEOPARD study team[¶]**

**1** Department of Pathology and Cell Biology, Columbia University Medical Center, New York, New York, United States of America, **2** Empilweni Services and Research Unit, Rahima Moosa Mother and Child Hospital, Department of Paediatrics and Child Health, Faculty of Health Sciences, University of the Witwatersrand, Johannesburg, South Africa, **3** Department of Biostatistics and Epidemiology, Rutgers School of Public Health, Piscataway, New Jersey, United States of America, **4** Department of Epidemiology, Mailman School of Public Health, Columbia University Medical Center, New York, New York, United States of America, **5** ICAP at Columbia University, Mailman School of Public Health, Columbia University Medical Center, New York, New York, United States of America, **6** Department of Pediatrics, Vagelos College of Physicians & Surgeons, Columbia University Medical Center, New York, New York, United States of America, **7** Centre for HIV and STIs, National Institute for Communicable Diseases, National Health Laboratory Services, and Faculty of Health Sciences, University of the Witwatersrand, Johannesburg, South Africa, **8** Gertrude H. Sergievsky Center, Vagelos College of Physicians and Surgeons, Columbia University Medical Center, New York, New York, United States of America

[¶]Membership of the EPIICAL Consortium and the LEOPARD study team is provided in the Acknowledgments.
* andrew.yates@columbia.edu

**Data Availability Statement:** Data are available at https://www.openicpsr.org/openicpsr/project/167981/version/V1/view.

## Abstract

In 2019 there were 490,000 children under five living with HIV. Understanding the dynamics of HIV suppression and rebound in this age group is crucial to optimizing treatment strategies and increasing the likelihood of infants achieving and sustaining viral suppression. Here we studied data from a cohort of 122 perinatally-infected infants who initiated antiretroviral treatment (ART) early after birth and were followed for up to four years. These data included longitudinal measurements of viral load (VL) and CD4 T cell numbers, together with information regarding treatment adherence. We previously showed that the dynamics of HIV decline in 53 of these infants who suppressed VL within one year were similar to those in adults. However, in extending our analysis to all 122 infants, we find that a deterministic model of HIV infection in adults cannot explain the full diversity in infant trajectories. We therefore adapt this model to include imperfect ART adherence and natural CD4 T cell decline and reconstitution processes in infants. We find that individual variation in both processes must be included to obtain the best fits. We also find that infants with faster rates of CD4 reconstitution on ART were more likely to experience resurgences in VL. Overall, our findings highlight the importance of combining mathematical modeling with clinical data to disentangle the role of natural immune processes and viral dynamics during HIV infection.

**Funding:** This work is part of the EPIICAL project
(http://www.epiical.org/), supported by the PENTA-
ID foundation (http://penta-id.org/), funded through
an independent grant by ViiV Healthcare UK. Data
were collected during the Latency and Early
Neonatal Provision of Antiretroviral Drugs Clinical
Trial (LEOPARD) study. The LEOPARD study was
supported in part by the Eunice Kennedy Shriver
National Institute of Child Health and Human
Development/National Institute of Allergy and
Infectious Disease, National Institutes of Health
(U01HD080441), USAID/PEPfAR, the South
African National HIV Programme, and South
African Research Chairs Initiative of the
Department of Science and Technology and
National Research Foundation of South Africa. The
funders had no role in study design, data collection
and analysis, decision to publish, or preparation of
the manuscript.

**Competing interests:** The authors have declared
that no competing interests exist.

## Author summary

For infants infected with HIV at or near birth, early and continued treatment with antiretroviral therapy (ART) can lead to sustained suppression of virus and a healthy immune system. However many treated infants experience viral rebound and associated depletion of CD4 T cells. Mathematical models can successfully capture the dynamics of HIV infection in treated adults, but many of the assumptions encoded in these models do not apply early in life. Here we study data from a cohort of HIV-positive infants exhibiting diverse trajectories in response to ART. We show that wide-ranging outcomes can be explained by a modified, but still remarkably simple, model that includes both the natural dynamics of their developing immune systems and variation in treatment adherence. Strikingly, we show that infants with strong rates of recovery of CD4 T cells while on ART may be most at risk of virus resurgence.

## Introduction

In 2019 there were 490,000 children under five living with HIV, and 150,000 newly diagnosed cases [1]. Although infants receiving antiretroviral treatment (ART) can suppress viral load (VL), eventually the cessation of treatment leads to HIV rebound, due to reactivation of latently-infected cells. Nevertheless, early initiation of ART can lead to extended periods of suppression in the absence of treatment—for example, over 22 months in the case of the 'Mississippi Child' and 8.75 years in a South African participant of the Children with HIV Early antiRetroviral therapy (CHER) trial [2, 3]. Therefore, understanding the dynamics of HIV suppression and rebound following ART initiation in young infants is crucial for optimizing treatment strategies and increasing the likelihood of achieving and sustaining viral suppression.

We previously showed that a simple biphasic model of VL decay captures the early dynamics of HIV decline in perinatally-infected infants on ART and that these dynamics are similar to those in adults [4]. However, models applied to dynamics of infection in adults over longer timescales typically encode assumptions that do not extend to infants [5–12]. First, CD4 T cell dynamics in adults are typically described as a balance between a constant total rate of influx and a constant *per capita* rate of loss, leading to steady trajectories in the absence of infection. In contrast, HIV-uninfected infants experience a natural, exponential decline in CD4 T cell numbers per unit volume of blood as the immune system matures [13]. Second, perinatally-infected infants undergo a transient period of CD4 T cell reconstitution upon ART initiation, during which numbers quickly recover to those of HIV-uninfected infants [14]. This short-lived process cannot be captured by the constant CD4 recruitment term exploited in many models of adult infection. Third, the standard assumption that ART is completely effective in blocking new infection of cells may not hold true for young infants, due to challenges in treatment adherence. Thus, canonical models of HIV suppression and rebound in adults must be modified for infants to include potential reductions in ART efficacy, and more complex dynamics of CD4 T cell numbers.

Here we model the dynamics of HIV infection in a cohort of perinatally-infected infants from Johannesburg, South Africa who initiated ART early in life. We extend a simple deterministic model of HIV suppression and rebound in adults to incorporate incomplete treatment adherence and dynamics of natural CD4 T cell decline and reconstitution. By fitting this model to longitudinal viral RNA and CD4 T cell data, we estimate rates of reactivation and reconstitution. We also show that individual variation in CD4 reconstitution rates and VL

persistence are important factors driving variation in HIV suppression and resurgence characteristics across infants, in addition to ART adherence. Overall, our results demonstrate the complex interplay between natural immune processes and HIV dynamics, and highlight the importance of mathematical modeling in disentangling these factors.

## Materials and methods

### Ethics statement

All protocols for the LEOPARD study were approved by the Institutional Review Boards of the University of the Witwatersrand and Columbia University. Written informed consent was obtained from mothers for their own and their infants' participation.

### Data

The LEOPARD study has been described previously [4, 15]. Briefly, 122 perinatally-infected infants were enrolled at the Rahima Moosa Mother and Child Hospital in Johannesburg, South Africa, between 2014 and 2017. The majority began ART within two weeks of birth (median age: 2.5 days; interquartile range (IQR): 1–8), and were followed for up to four years. VL (HIV RNA copies ml$^{-1}$) and CD4 T cell concentrations (cells $\mu$l$^{-1}$) in the blood were sampled over time, and various clinical covariates were also recorded, including the infant's pre-treatment CD4 percentage, the mother's VL and CD4 count after delivery, and the mother's prenatal ART history (S1(A) Text).

With these data, we previously identified a subset of 53 infants who successfully suppressed VL within one year [4, 16], with suppression defined as having at least one VL measurement below the 20 copies ml$^{-1}$ detection threshold of the RNA assay. Here, we are interested in the interplay between natural CD4 T cell dynamics and infection processes, and whether and how this interplay determines whether an infant achieves suppression and/or experiences VL rebound. We have therefore broadened our analysis to all 122 infants.

In addition to the data described previously, we include information relating to ART adherence that was obtained at each study visit (S1(B) Text and S1 Fig). For each drug in each infant's ART regimen—with the recommended, and most common, being zidovudine (AZT), lamivudine (3TC), and (i) nevirapine (NVP) in the first four weeks of treatment, then (ii) ritonavir-boosted lopinavir (LPV/r) after four weeks—we estimated a percentage adherence by comparing the weight of medicine returned to the expected amount returned assuming perfect adherence. Less than 100% adherence can result from missed doses or 'under-dosing' (giving too little medicine at each dose), whereas greater than 100% adherence can occur through over-dosing or problems with drug tolerance (infants may spit up bad-tasting medicine, therefore requiring repeat dosing). For many visits, adherence could not be calculated because leftover medicine was not returned. The adherence estimates are therefore influenced by many unobserved factors and, given uncertainty in how the quantitative estimates map to actual adherence, we instead defined a categorical variable that labeled adherence estimates greater than 90% as 'good', and estimates less than 90% as 'poor'. Using 85% and 95% as alternative thresholds for good adherence did not alter our findings. With this course-grained approach, some missing values could be manually labelled based on physician commentary from accompanying questionnaires (for example, if substantial gaps in dosing were noted, adherence was labeled as poor). We then summarized the average adherence of each infant as the most frequently reported category (good or poor) across their time series.

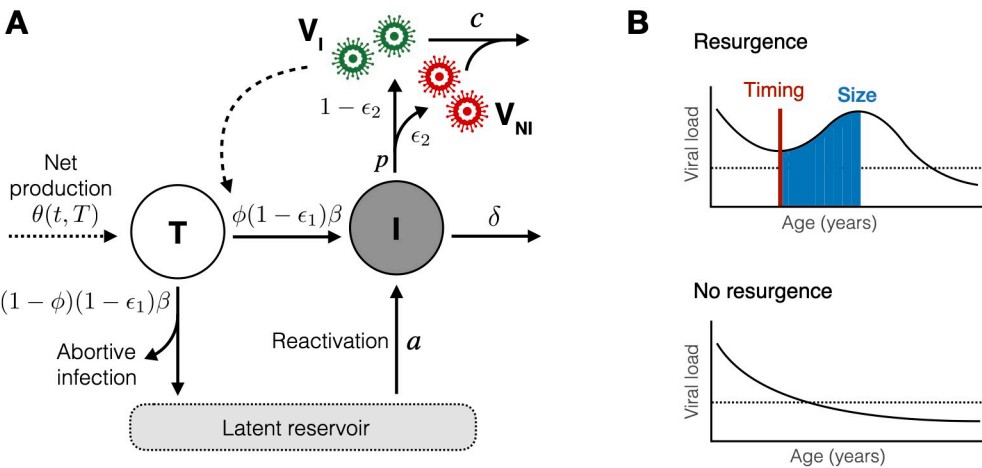

**Fig 1. Model framework and analysis schematic.** (A) The infection model with rate constants. CD4 target cells (T) are infected by free infectious virus ($V_I$) and either become productively infected cells (I), die though abortive infection, or become latently infected. Productively infected cells produce both infectious (green) and non-infectious (red) virus, both of which are cleared at rate $c$, and latently infected cells can become reactivated at a later point to join the productively infected cell population. CD4 target cells also undergo reconstitution and natural decline processes at total net rate $\theta(t, T)$. Further details are given in the text. (B) Schematic illustrating the definition of viral resurgence (top) compared to no resurgence (bottom). The timing of resurgence is defined as the time at which viral load first starts increasing (vertical red line), and the size of resurgence is the total integrated viral load during the upslope period (blue shaded region). The dashed horizontal line represents the detection threshold of the assay.

## Model

We describe HIV dynamics in an infant on ART using a deterministic ordinary differential equation (ODE) model (Fig 1A) [7, 12]. We assume that CD4 target cells, $T(t)$ (measured as the concentration of cells per $\mu$l of blood, but from here on referred to as 'cells' or 'counts' for brevity), undergo background growth and loss at net rate $\theta(t, T)$, and are infected by free infectious virus, $V_I(t)$, with a *per capita* transmission rate $\beta$ (Table 1). During ART this transmission

**Table 1. Model parameters and population-level estimates.**

| Parameter | Description | Units | Value, if fixed | Mean (SE), if estimated[*] |
|---|---|---|---|---|
| $\beta_0 = (1 - \epsilon_1)(1 - \epsilon_2)\beta$ | Per-cell effective transmission rate | (copies ml$^{-1}$)$^{-1}$ day$^{-1}$ | | $4.0 \times 10^{-8}$ ($9.5 \times 10^{-9}$) |
| $\phi$ | Proportion of infections that are productive | – | 0.05–0.35[†] [17, 18] | |
| $\bar{p} = p/c$ | Ratio of viral production to loss | copies ml$^{-1}$ cell$^{-1}$ | | 1210 (174) |
| $\delta$ | Average rate of loss of infected cells | day$^{-1}$ | | 0.06 (0.007) |
| $a$ | Total rate of latent reservoir reactivation | cells day$^{-1}$ | | $2.1 \times 10^{-4}$ ($5.7 \times 10^{-5}$) |
| $b_0$ | Extent of natural CD4 T cell decline | cells | -2354 [13] | |
| $-1/b_1$ | Timescale of natural CD4 T cell decline | days | 1003 [13] | |
| $r$ | Rate of CD4 T cell reconstitution | cells day$^{-1}$ | | 7.5 (0.5) |
| $T_R$ | Age at reconstitution plateau | days | 210–235[†][14] | |
| $T_0$ | Initial number of CD4 T cells | cells | | 1659 (76) |
| $V_0$ | Initial viral load | copies ml$^{-1}$ | | 7386 (1838) |

[*]Estimates taken from the model with lowest AIC

[†]A range of values around previous estimates was explored

SE = standard error of the fixed effect; cells = cells $\mu$l$^{-1}$

is blocked with efficacy $\epsilon_1$, where $\epsilon_1 < 1$ reflects incomplete adherence to reverse transcriptase inhibitors (for example, NVP and AZT) and/or their failure to completely block infection of new cells. A proportion $(1—\phi)$ of the newly infected target cells either die through abortive infection [17, 18] or seed the latent reservoir; the remaining fraction, $\phi$, become productively infected. These infected cells $I(t)$ are lost at an average rate $\delta$, and produce infectious virus at an average *per capita* rate $(1 - \epsilon_2)p$, where $\epsilon_2 < 1$ reflects incomplete adherence to protease inhibitors (for example, LPV/r) and/or incomplete blocking of the production of infectious virus. The infected cell population is also boosted by reactivation of the latent reservoir, at total rate $a$. We do not explicitly model the number of latently infected cells due to uncertainties in the rates of proliferation and loss in this population, and a lack of available data to estimate these parameters. Finally, free virus is lost at rate $c$. This system is represented by the following equations;

$$\frac{dT}{dt} = \theta(t, T) - (1 - \epsilon_1)\beta V_I T$$

$$\frac{dI}{dt} = \phi(1 - \epsilon_1)\beta V_I T - \delta I + a$$

$$\frac{dV_I}{dt} = (1 - \epsilon_2)pI - cV_I$$

$$\frac{dV_{NI}}{dt} = \epsilon_2 pI - cV_{NI}.$$

The basic reproduction number for this model at ART initiation is

$$R_0 = \frac{\phi(1 - \epsilon_1)(1 - \epsilon_2)\beta p T_0}{c\delta},$$

where $T_0$ is the initial CD4 count, and we assume the contribution of the latent reservoir to the production of infected cells is negligible at this time.

In the simplest case we assume all rate parameters are constant over time. We also investigated an extension of this model that incorporates a delay in reactivation of the latent reservoir, such that

$$a = a(t) = \begin{cases} 0 \text{ if } t \leq T_A \\ \bar{a} \text{ if } t > T_A, \end{cases}$$

where $T_A$ is the time to reactivation in days. Assuming the rate of virus turnover is faster than that of CD4 T cells [19], we reduce the model to the following system (S1(C) Text);

$$\frac{dT}{dt} = \theta(t, T) - \beta_0 VT$$

$$\frac{dV}{dt} = \phi \bar{p}\beta_0 VT - \delta V + a\bar{p},$$

where $V$ is total free virus $V_I + V_{NI}$, $\beta_0 = (1 - \epsilon_1)(1 - \epsilon_2)\beta$, and $\bar{p} = p/c$. The compound parameter $\bar{p}$ represents the contribution of each infected cell to the total viral load, through its rate of production of both infectious and non-infectious virions ($p$) and the average time they persist in circulation ($1/c$). The compound parameter $\beta_0$ incorporates both transmission and the total efficiency of both modes of action of ART (through $\epsilon_1$ and $\epsilon_2$), and thus includes the extent of each infant's adherence to treatment. In this model, $R_0 = \phi\beta_0 T_0\bar{p}/\delta$. Note that because $\beta_0$ is a single parameter which we estimate from the data, we cannot use this expression to estimate the pre-treatment value of $R_0$.

Finally, we extend the model for young infants through the term governing background CD4 T cell growth and loss, $\theta(t, T)$. In models of infection dynamics in adults, $\theta(t, T)$ typically takes the form $\lambda - d_T T$, where $\lambda$ and $d_T$ are constant rates representing cell influx and natural decay processes, respectively. These forms lead to steady trajectories in the absence of infection. For infants, we propose an alternative $\theta(t, T)$ that instead accounts for (i) the exponentially declining concentration of CD4 T cells that is observed as HIV-uninfected infants age [13], and (ii) the transient recovery in CD4 counts experienced by HIV-infected infants during the early stages of ART [14]. First, the natural decline in CD4 T cells can be captured by an exponential decay function

$$T = c_0 + b_0(1 - e^{b_1 t}),$$

where $c_0$, $b_0$ and $b_1$ are constant parameters that have been independently estimated in a cohort of 80 uninfected children in Germany, including 39 aged between 2 months and 4 years [13]. This function also captured CD4 T cell dynamics in 381 South African children, of whom 300 were aged between 2 weeks and 5 years [20]. Second, the additional reconstitution of the CD4 T cell pool in HIV-infected infants can be modeled as a transient increase in cell counts during the early stages of ART, *i.e.*

$$\frac{dT}{dt} = \begin{cases} r \text{ if } t \leq T_R \\ 0 \text{ if } t > T_R, \end{cases}$$

where $r$ is the constant rate of reconstitution and $T_R$ is the time to reach healthy levels [14]. Combining these processes of reconstitution and the natural decline of CD4 T cell counts in infants gives $\theta(t, T) = -b_1 b_0 e^{b_1 t} + \bar{r}$, and

$$\frac{dT}{dt} = -b_1 b_0 e^{b_1 t} + \bar{r} - \beta_0 VT \tag{1}$$

$$\frac{dV}{dt} = \phi \bar{p} \beta_0 VT - \delta V + a\bar{p}, \tag{2}$$

where

$$\bar{r} = \begin{cases} r \text{ if } t \leq T_R \\ 0 \text{ if } t > T_R. \end{cases}$$

## Model fitting and comparisons

We fit Eqs 1 and 2 to the VL and CD4 T cell data from all 122 infants using a nonlinear mixed effects approach. All VL observations below the detection threshold were treated as censored values, and we assumed both $V(t)$ and $T(t)$ were lognormally distributed [21, 22]. Given the relative infrequency of CD4 T cell measurements, we fixed four parameters across all individuals (Table 1): three that governed the reconstitution and natural dynamics of target cells ($T_R$, $b_0$ and $b_1$), and the proportion of newly infected cells that become productively infected ($\phi$). All other parameters were estimated and allowed to have both fixed and random effects. In subsequent analyses we estimated fixed and random effects for $T_R$ and $\phi$. We also examined the importance of individual variation in adherence, VL persistence and CD4 T cell recovery by comparing the best fit model to three alternative models in which $\beta_0$, $\bar{p}$ or $r$ were fixed across all infants.

Following exploratory fits, each estimated parameter was assumed to follow a lognormal distribution, with the exception of $a$ which followed a logit-normal distribution with pre-specified upper bound, and $T_R$ which followed a normal distribution. We verified that the random effects for all estimated parameters were normally distributed, using the Shapiro-Wilk test. Guided by the exploratory fits, we allowed $\beta_0$ and $d$ to be correlated, and assumed all other parameters were independent. We confirmed the structural identifiability of all parameters [23], detailed in S1(D) Text, and conducted additional sensitivity analyses by varying each chosen parameter in turn and re-simulating the model, while keeping all other parameters fixed. We used these simulations to assess the sensitivity of model predictions to our choice of fixed parameters. Model fitting and parameter estimation were implemented in Monolix 2020R1 [22], detailed in S1(E) Text. Downstream analyses and plotting were conducted in R version 4.03 [24], with the `deSolve, cowplot, patchwork` and `tidyverse` packages [25–28].

We compared the statistical support for different models using the Akaike Information Criterion (AIC). For model $i$, $AIC_i = 2k - 2 \ln L$, where $k$ is the number of estimated parameters, $\ln L$ is the maximum log-likelihood, and lower AIC values indicate stronger statistical support. We assessed the relative support for model $i$ using $\Delta AIC_i = AIC_i - AIC_{min}$, where $AIC_{min}$ is the minimum AIC value across all models. Differences greater than five indicate substantially greater support for the model with $AIC_i = AIC_{min}$. For the favored model, we used the individual-specific parameter estimates to predict VL and CD4 T cell trajectories for each child. These trajectories extended either to the end of our study period or two years after their last observation, whichever was earlier. We then compared how viral infection and the natural decline in CD4 T cells mediated the overall VL and CD4 T cell dynamics. We calculated the relative contributions of new viral infection and natural decline to decreases in CD4 T cell numbers as

$$\frac{\beta_0 TV}{\beta_0 TV + b_1 b_0 e^{b_1 t}} \qquad \text{and} \qquad \frac{b_1 b_0 e^{b_1 t}}{\beta_0 TV + b_1 b_0 e^{b_1 t}}, \tag{3}$$

respectively. Similarly, the relative contributions of new viral infection and latent reservoir reactivation to increases in the number of productively infected cells were

$$\frac{\phi \beta_0 TV}{\phi \beta_0 TV + a} \qquad \text{and} \qquad \frac{a}{\phi \beta_0 TV + a}, \tag{4}$$

respectively.

## Statistical analyses

We tested for statistical associations between model parameters, clinical covariates (S1(A) Text), and the risk of VL resurgence—defined as any predicted increase in VL following initiation of ART (Fig 1B). We chose VL resurgence as our indicator of imperfect viral control rather than VL rebound (any predicted increase in VL following initial suppression of HIV) due to the small number of infants experiencing the latter (7/122 infants experienced rebound compared to 53/122 experiencing resurgence). We defined the timing of VL resurgence as the first point at which the model predicted an increase in VL, and the size of resurgence as the total integrated VL during the upslope period (Fig 1B). We then tested for associations using Spearman correlations between pairs of quantitative variables, the Kruskal Wallis test between quantitative and categorical variables, and Chi-squared tests between pairs of categorical variables. We adjusted for multiple testing using the Benjamini-Hochberg correction.

## Results

The model for adult infection, with $\theta(t, T) = \lambda - d_T T$, was a poor fit to the infant data, particularly the CD4 T cell counts (S2 Fig, Table 2). We therefore used the model adapted for infant infection, with $\theta(t, T) = -b_1 b_0 e^{b_1 t} + \bar{r}$, in all further analyses. First, we verified that the infant model with fixed time to reconstitution plateau, $T_R$, and constant rate of latent reactivation, $a$, was structurally identifiable (see Table 1 and S1(D) Text) [23]. We initially fixed $T_R = 222$ days across all infants, following previous modeling of CD4 reconstitution in another cohort of HIV-infected infants who initiated ART 82 days after birth, on average [14]. We refitted the model with different fixed values and verified that the best fits were obtained when $T_R = 230$ days (S3 Fig). We also varied $\phi$ between 0.05 and 0.35 [17, 18] and found the best fit for $\phi = 0.35$. Including random effects for $T_R$ and $\phi$ did not improve model fits, nor did estimating the fixed effects (*i.e.*, estimating the population average of $T_R$ and $\phi$; S1(F) Text). This is likely due to the increased complexity introduced by estimating these additional parameters. Similarly, including a delay in reactivation of the latent reservoir did not improve model fits (Table 2). We therefore focus on the model with a constant rate of reactivation, fixed $T_R = 230$ days, and fixed $\phi = 0.35$. With this model, VL predictions were marginally sensitive to $T_R$ and $\phi$ (S4 Fig), whereas CD4 T cell dynamics were only sensitive to $T_R$ (S5 Fig).

Strikingly, our relatively simple deterministic model captured the wide variation in infant VL trajectories, including monotonic decreases to suppression, eventual suppression following transient increases in VL, and brief periods of suppression with a subsequent rebound in VL (Fig 2). Later, or multiple, rebound occurrences were generally not so well captured. These behaviors may be due to repeated fluctuations in treatment adherence or stochastic processes driving delayed reactivation of the latent reservoir, neither of which are included in the model. Initially, new infections were the major contributor to growth of the productively infected cell population (Eq 4; S6 Fig). However, in almost all infants the importance of new infection events was eventually superseded by reactivation from the latent reservoir, although this displacement was delayed by viral resurgence events (S7 Fig).

The majority of infants experienced a transient increase in CD4 T cell counts followed by a steady decline; these patterns were well captured by the model (Fig 3). The decline in CD4 T cells was almost always driven by natural processes, although the contribution of new infections increased during periods of VL resurgence (Eq 3; S8 Fig).

The fixed effects for all estimated parameters, and the standard error of the fixed effects, are given in Table 1. The average lifespan of productively infected cells ($1/\delta$) across all individual infant estimates was 16 days (2.5%—97.5% percentile range = 3–61 days), and $R_0$ at ART initiation was 0.48 (0.29–1.00), reflecting an initial decrease in VL across most infants. The rate of CD4 T cell reconstitution, $r$, was positively correlated with the initial number of CD4 T cells,

**Table 2. Model comparisons.** AIC values (ΔAIC) are quoted relative to the minimum AIC value across all models. The model with ΔAIC = 0 is the model with lowest AIC and thus has most statistical support. See Table 1 for parameter definitions.

| Model* | ΔAIC |
|---|:---:|
| Constant reactivation, $a$ | 0.0 |
| Time-varying reactivation, $a = a(t)$ | 212.9 |
| $\theta(t, T) = \lambda - d_T T$ | 373.4 |
| No reactivation, $a = 0$ | 419.3 |

*Unless stated otherwise, $\theta(t, T) = -b_1 b_0 e^{b_1 t} + \bar{r}$ with $T_R = 230$ days and $\phi = 0.35$.

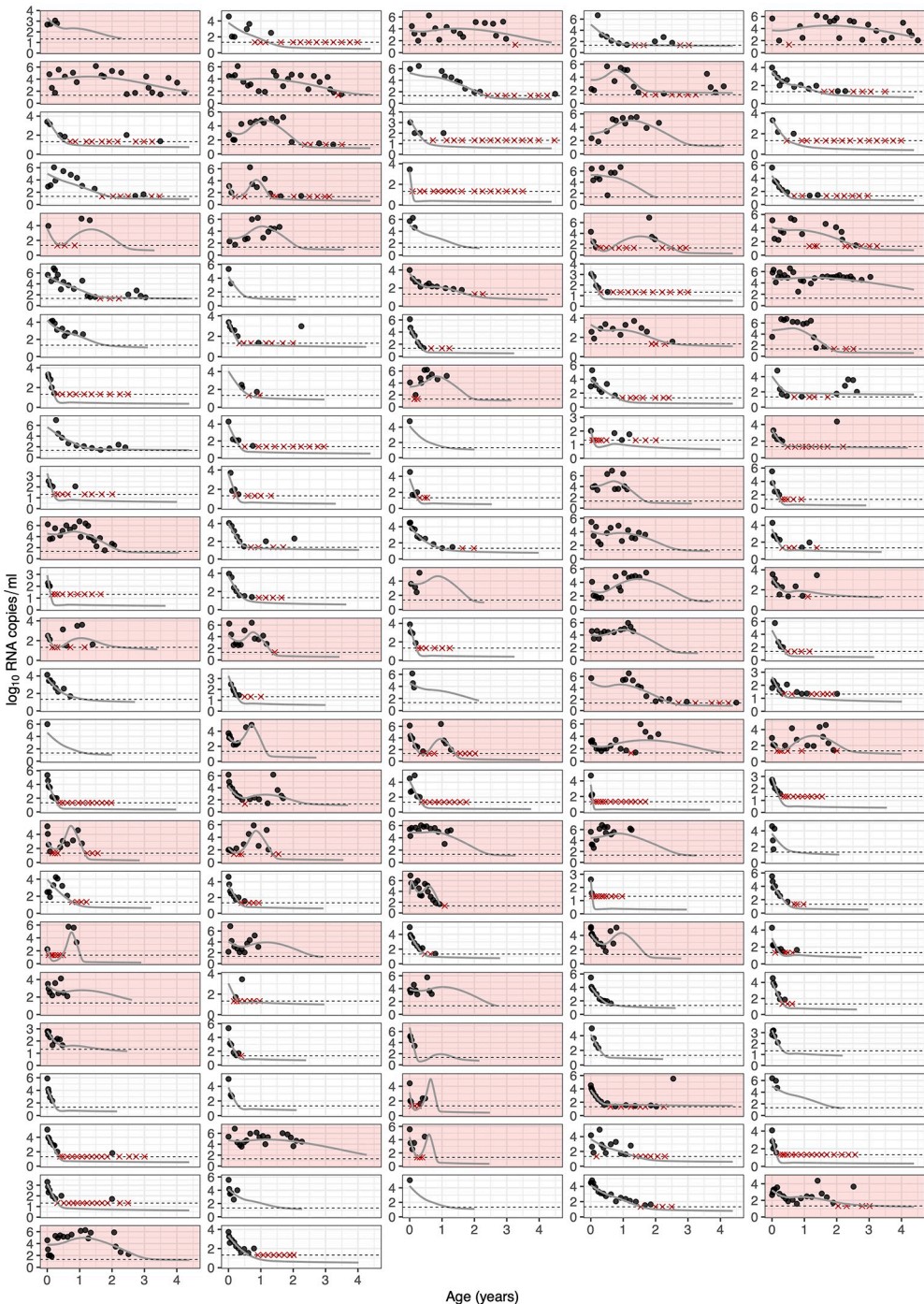

**Fig 2. Model fits for viral RNA observations.** Each panel represents a different infant; points represent the data; and solid lines are the model fits. The dashed horizontal line is the detection threshold of the RNA assay, and red crosses are censored observations below this threshold. Panels shaded in red are infants who experienced viral resurgence (i.e. at least one period of increasing VL).

$T_0$ (S9 Fig), in contrast to the negative correlation reported elsewhere in other young cohorts [14, 29]. The basis of this difference is unclear, although we note that the infants in the LEOP-ARD cohort all initiated ART at a much younger age than the children in those studies. This association is unlikely to be driven by poor parameter identifiability, which would instead

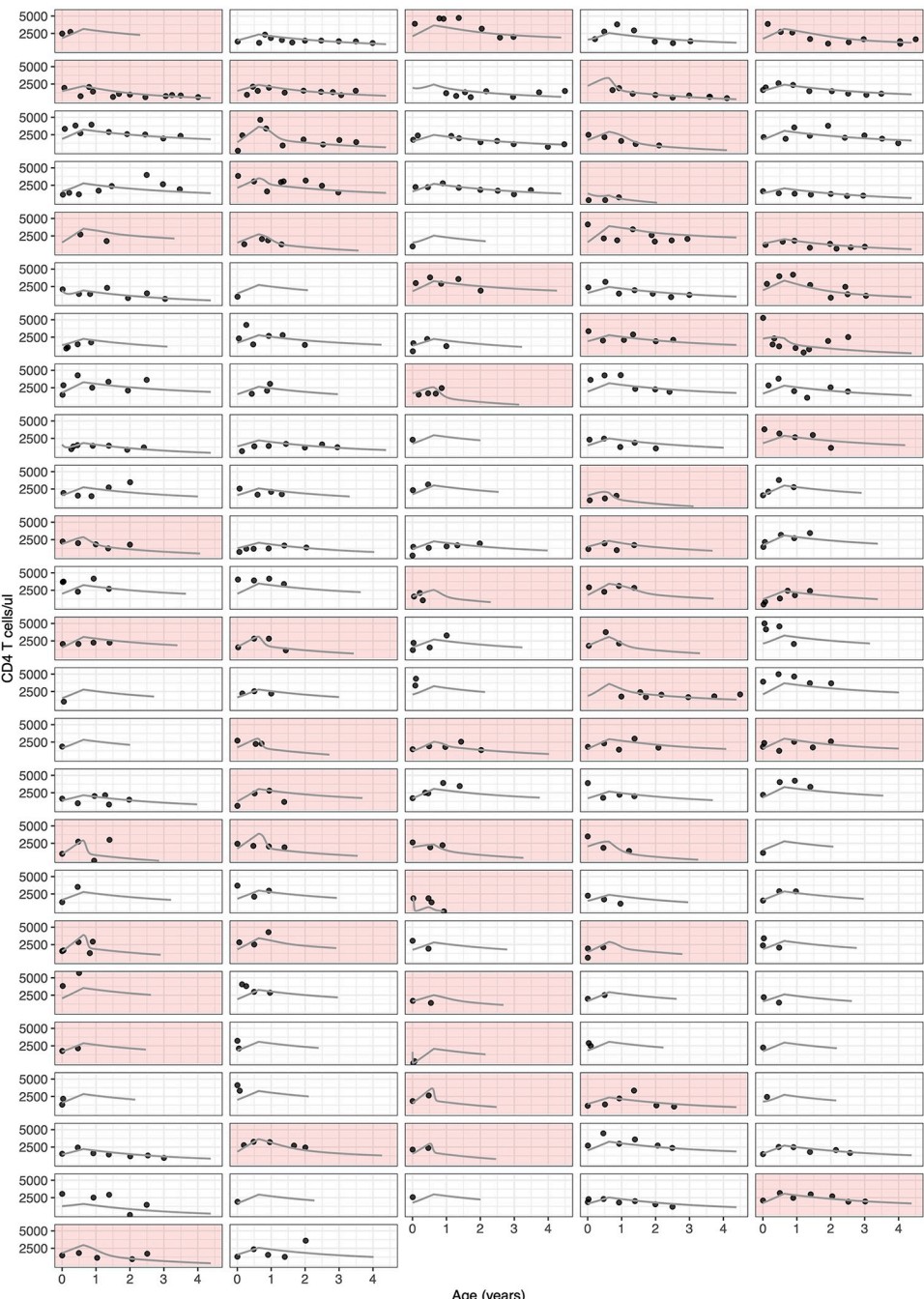

**Fig 3. Model fits for CD4 T cell observations.** Each panel represents a different infant, ordered as in Fig 2; points represent the data; and solid lines are the model fits.

cause negative correlations through compensatory mechanisms. Notably, we found that infants with higher reconstitution rates, $r$, and higher VL production to decay ratios, $\bar{p}$, were more likely to experience increases in VL after ART initiation ($p < 1 \times 10^{-6}$; Fig 4A and 4B). For those infants who did experience increases in VL, larger and earlier increases were associated with higher VL production to decay ratios ($p < 10^{-4}$; Fig 4C and 4D), but not reconstitution rates ($p > 0.1$). Overall, while it is intuitive that rebound dynamics would be positively

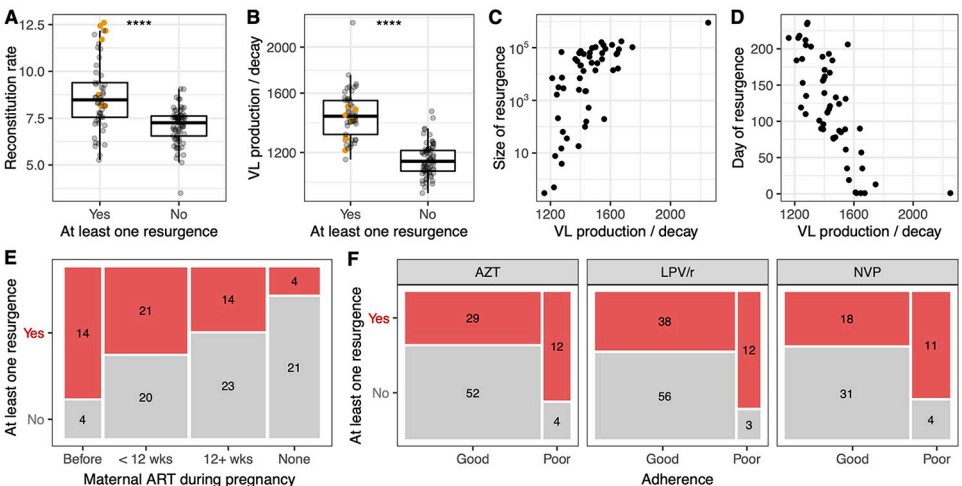

**Fig 4. VL resurgence is associated with rates of CD4 reconstitution, VL production and decay, and ART history of infant and mother.** (A–B) Relationship between the occurrence of VL resurgence (defined as any increase in VL following initiation of ART) and the CD4 reconstitution rate, $r$, in cells $\mu l^{-1}$ day$^{-1}$ (A) and the ratio of virus production to decay, $\bar{p} = p/c$, in copies ml$^{-1}$ cell$^{-1}$ (B). Each point represents a different infant and $p < 0.0001(****)$ in both cases. Seven infants whose resurgence was a viral rebound event are highlighted in orange. (C–D) Relationship between $\bar{p}$ and the size of VL resurgence in RNA copies ml$^{-1}$ (C) and timing of VL resurgence in days (D). Each point represents an infant who experienced resurgence. Correlations are 0.66 and -0.75, respectively, and $p < 0.0001$ in both cases. (E) Relationship between the occurrence of VL resurgence and the timing of maternal ART initiation ($p < 0.01$). The size of each box reflects the proportion of infants in the corresponding category and the numbers show the corresponding sample size. (F) Relationship between infant ART adherence and the occurrence of VL resurgence ($p < 0.05$ for AZT and LVP/r; $p = 0.06$ for NVP). Adherence was classified as 'good' if the majority of adherence estimates were 90% or more, and 'poor' otherwise.

associated with $\bar{p}$, which reflects both the average rate of virus production by infected cells and the persistence time of free virus in circulation, we find that the CD4 reconstitution rate $r$ is also a key parameter determining an infant's propensity for resurgence events.

In addition to the model parameters, we found a longer duration of maternal prenatal ART was associated with risk of VL resurgence ($p < 0.01$; Fig 4E), as were poor LVP/r and AZT adherence ($p < 0.05$; Fig 4F). However, maternal ART and our binary adherence covariates were also associated with higher VL production to decay ratios (S10 Fig; all $p < 0.01$), suggesting potential colinearity. All other associations between VL resurgence characteristics and clinical covariates, including pre-treatment CD4 percentage and age at ART initiation (S1(A) Text), were not significant at the $\alpha = 0.05$ level.

## Variation in viral persistence, reported adherence, and the natural dynamics of CD4 T cells dictate infant trajectories

The most obvious explanation for the wide variety of VL trajectories we have identified here is variation in ART adherence, which may be more pronounced in infants than adults. Indeed, we found that reported poor adherences to LVP/r or AZT, which are the treatments most commonly administered long-term, were associated with resurgence (Fig 4F). In the model, adherence is reflected in the parameters $\epsilon_1$ and/or $\epsilon_2$, which dictate the efficiency of treatment at blocking new infection and virus production by infected cells, respectively. These parameters were not individually identifiable with these data, but instead are subsumed in the compound parameter $\beta_0 = (1 - \epsilon_1)(1 - \epsilon_2)\beta$. If adherence were the main driver of variation in VL trajectories, then inter-individual variation in $\beta_0$ should be the most crucial component of our model. Puzzlingly, however, we found no association between $\beta_0$ and any measure of resurgence, or

**Table 3. Model comparisons of adherence and CD4 recovery parameters.** AIC values (ΔAIC) are quoted relative to the minimum AIC value across all models. The model with ΔAIC = 0 is the model with lowest AIC and thus has most statistical support. See Table 1 for parameter definitions.

| Model | ΔAIC |
|---|---|
| Fixed and random effects for $\beta_0$, $\bar{p}$ and $r$ * | 0.0 |
| Fixed and random effects for $\beta_0$ and $\bar{p}$; only fixed effects for $r$ | 44.5 |
| Fixed and random effects for $\beta_0$ and $r$; only fixed effects for $\bar{p}$ | 53.1 |
| Fixed and random effects for $\bar{p}$ and $r$; only fixed effects for $\beta_0$ | 358.8 |

*Corresponds to the best-fit model in Table 2.

between $\beta_0$ and average LVP/r or AZT adherence ($p>0.05$ in all cases), although we did find a negative association between $\beta_0$ and adherence to the early-phase treatment NVP ($p<0.05$). Instead, within the model we found $r$ and $\bar{p}$ to be the key predictors of resurgence (Fig 4A–4D), suggesting that variation in CD4 T cell and VL persistence dynamics are important. We speculate that the strong correlation between $\beta_0$ and the infected cell death rate $\delta$ across children (S9 Fig) masks the effect of inter-individual variation in the parameters $\epsilon_1$ and/or $\epsilon_2$, which is better captured by the adherence data derived from questionnaires. To explore this issue, we refit the model while removing the random effects for $\beta_0$, $\bar{p}$ and $r$ in turn. Fixing any of these parameters resulted in substantially poorer fits ($\Delta$AIC > 40; Table 3), suggesting that variation across individuals in CD4 T cell dynamics, VL persistence, and ART adherence all drive variation in HIV suppression and resurgence characteristics across infants.

## Discussion

In this study we modeled the dynamics of HIV suppression and rebound in perinatally-infected infants receiving ART. Our framework extends previous models of rebound in adults [7, 12] by incorporating mechanisms of the natural decline and infection-induced reconstitution of CD4 T cells in young infants [14]. We found that new infection events were initially the major contributor to growth of the productively infected cell population, but that reactivation of the latent reservoir became more important once VL levels were low. We also identified natural processes as the long-term driver of declining CD4 T cell frequencies in blood. What was perhaps unexpected was that our simple framework can capture large variations in infant VL trajectories, including monotonic decreases to sustained suppression, resurgences in VL, and suppression with subsequent rebound. Although the canonical explanation for erratic VL patterns is imperfect ART adherence, we found that incorporating variation in CD4 reconstitution rates was also required to capture the complexity in our infant data. We demonstrate that within a deterministic framework the interplay of natural CD4 dynamics, constant levels of latent reservoir reactivation, and constant ART efficacy can recapitulate intricate infection dynamics. Our analyses indicate that with the levels of adherence achieved in this study, resurgence may in fact be inevitable for infants with certain virological and CD4 T cell parameter combinations.

Although our estimates of the average CD4 T cell reconstitution rate ($r$ = 7.5 cells $\mu$l$^{-1}$ day$^{-1}$) is greater than those from another cohort of HIV-infected infants ($r$ = 3.8 cells $\mu$l$^{-1}$ day$^{-1}$; ref. [14]), it is within the interquartile range. Notably, infants from this other cohort initiated ART later, on average, than the infants in our cohort (median = 82 days, 25th percentile = 34, 75th percentile = 121), and all eventually achieved viral suppression. We also found that higher rates of reconstitution were associated with a greater probability of experiencing a resurgence

in VL. This relationship was not confounded by the immunological status of infants at the beginning of the study as we found no association between the reconstitution rate and pre-treatment CD4 percentage or counts, or between the risk of VL resurgence and pre-treatment CD4 percentage or counts. Our finding raises the possibility that rapid recovery of CD4 T cells, despite suggesting an improved clinical state, can also increase the risk of VL resurgence in some individuals by repopulating the target cell pool. Although it could also be that VL resurgence triggers more rapid CD4 reconstitution through increased anti-viral immune activity or density-dependent responses to CD4 depletion [14], the latter seems unlikely in this cohort as we did not detect a negative association between the initial number of CD4 T cells ($T_0$) and $r$. Nevertheless, further investigation is needed to determine the directionality of this relationship, and whether the extent of CD4 T cell recovery may be used as a biomarker for individuals at increased risk of VL resurgence.

Our estimate of another key parameter, the rate of latent cell reactivation ($a = 2 \times 10^{-4}$ cells $\mu$l$^{-1}$ day$^{-1}$), is within the range of estimates obtained from adults during ART interruption ($2 \times 10^{-6}$—$1 \times 10^{-3}$ cells $\mu$l$^{-1}$ day$^{-1}$[7]). Biologically, a higher burden of reactivation ($a$) may reflect a larger latent reservoir in these infants and/or an increased per-cell rate of latent cell reactivation. Dynamically, larger reactivation estimates may compensate for significant fluctuations in treatment adherence that are not included in the model. We did not find any associations between $a$ and the occurrence or size of VL resurgence. This is perhaps not surprising as $a$ effectively represents the total contribution of latent cell reactivation averaged over the entire study, and its contribution to changes in VL relative to those of *de novo* infection events soon after ART initiation tends to be small (S6 Fig).

One counterintuitive result is that longer durations of maternal prenatal ART were associated with VL resurgence. This result could not be explained by worse adherence in this group ($p > 0.2$). However, we could not disentangle the effects of this variable from that of VL production to decay ratios, $\bar{p}$. Another study of this cohort found that longer exposure to maternal prenatal ART is associated with a larger viral reservoir [30]. It was speculated that maternal ART could lead to a larger representation of infants who acquired infection earlier during the pregnancy, potentially before ART was initiated, and hence have had a longer time to progress. Alternatively, there may be an enrichment of immuno-genetic risk factors in infants who become infected despite maternal ART.

There are a number of caveats to our modeling approach. First, our model does not differentiate between short- and long-lived productively infected cells, the loss of which underpin the multiphasic decline of VL in adults and infants on ART [4, 7, 31–34]. Instead, our estimate of the mean lifespan of a productively infected cell ($1/\delta$) is effectively a weighted average of the mean lifespans of all productively-infected subpopulations. Our estimate (16 days) is consistent with the median lifespan of 17 days we estimated previously from the VL dynamics in a subset of these infants who achieved suppression [4], and roughly in line with a recent estimate of the loss rate of infected CD4 T cells with intact proviruses [35]. Second, we do not explicitly model the dynamics of latently infected cells as they are not directly observed. Instead, the parameter $a$ in our model is effectively a 'force of reactivation', which combines the effects of reservoir size and the per-cell rate of reactivation. Third, we fit the peripheral CD4 T cell data to the number of target cells predicted by the model ($T(t)$), rather than the predicted sum of target cells, productively infected cells and latently infected cells. This approach is reasonable as the frequency of infection among CD4 T cells is typically small (S11 Fig and ref. [36]), and the majority of infected cells most likely reside in lymphoid tissues where infection-induced depletion of CD4 T cells is greatest [37]. We also assume all CD4 T cells are equally susceptible to infection, although in reality activated cells may be more susceptible than resting cells [38, 39]. However, this heterogeneity is implicitly incorporated within the transmission parameter,

$\beta$, if the proportion of CD4 T cells that are susceptible remains approximately constant over time.

Finally, we acknowledge that the ART regimens used in the LEOPARD trial may not be optimal. Although considered most effective at the time of study design and implementation, more potent treatments—for example, integrase inhibitors and/or broadly neutralizing antibodies—have since been approved for young infants. It will be important to determine whether infants starting these newer treatments are also at risk of resurgence, as we have identified here.

In conclusion, we have extended the canonical framework for HIV suppression and rebound to include more realistic dynamics of CD4 T cell decline and reconstitution in young infants on ART. We estimated rates of reactivation and reconstitution, and identified distinct phases in which dynamics were either dominated by new infection of CD4 T cells, or by reactivation of the latent reservoir. We also demonstrated the importance of incorporating variation in CD4 reconstitution rates to capture the diversity of infant VL trajectories. Overall, our results suggest that VL resurgence in perinatally-infected infants may be inevitable in certain parameter regimes, and highlight the utility of mathematical modeling in understanding the dynamics of infant HIV infection.

## Supporting information

**S1 Text. Descriptions of clinical covariates and assessments of infant adherence, the derivation of the model, its structural identifiability, and our approach to parameter estimation using *Monolix*.** A—List of clinical covariates. B—Additional information regarding infant adherence. C—Model. D—Structural Identifiability. E—Nonlinear Mixed Effects Modeling in Monolix.
(PDF)

**S1 Fig. Reported adherence trajectories.** Adherence estimates greater than 90% were labeled 'good'; and all others 'poor'. Each panel represents a different infant.
(PDF)

**S2 Fig. The standard model for adult CD4 T cell dynamics does not capture infant data.** Each panel represents a different infant, points represent the data, and solid lines are the model fits. Here $\theta(t, T) = \lambda - d_T T$, with $\lambda$ and $d_T$ assumed to have lognormal distributions. Initial estimates for the population mean were 1000 cells $\mu l^{-1}$ day$^{-1}$ and 0.25 day$^{-1}$, respectively, and for the standard deviation were 1 and 0.1, respectively.
(PDF)

**S3 Fig. Comparing models with different fixed values of $T_R$ and $\phi$ across all infants.** The AIC difference for model $i$ was calculated as AIC$_i$– AIC$_{min}$, where AIC$_{min}$ is the minimum AIC value across all models. The model with zero difference is the model with lowest AIC and thus is the most strongly favored.
(PDF)

**S4 Fig. Sensitivity of VL predictions to model parameters.** Each fixed (A) or estimated (B) parameter was varied within 20% of its original value while keeping all other parameters at their original values. Original values for the estimated parameters were the population-level means from the best-fit model.
(PDF)

**S5 Fig. Sensitivity of CD4 T cell predictions to model parameters.** Each fixed (A) or estimated (B) parameter was varied within 20% of its original value while keeping all other

parameters at their original values. Original values for the estimated parameters were the population-level means from the best-fit model.
(PDF)

**S6 Fig. Relative contributions of new infection events (grey) and LR reactivation (blue) to the generation of productively infected cells.** Each panel represents an infant, and red shaded regions show their VL scaled by its maximum value.
(PDF)

**S7 Fig. VL and CD4 T cell dynamics influence the time at which reactivation contributes most to productively infected cell growth.** Each point represents a different infant with respect to the time at which reactivation became the major contributor to productively infected cell growth and the time at which: (A) their VL started increasing (if applicable) and (B) their VL finished increasing (if applicable); $p = 0.6$ and $p < 0.001$, respectively, and the Spearman's rank correlation coefficient for (B) is 0.61.
(PDF)

**S8 Fig. Relative contributions of healthy dynamics (green) and depletion through infection (grey) to the decline in CD4 T cell densities in blood.** Each panel represents an infant, and red shaded regions show the periods of increasing VL (from start to peak, as shown in Fig 1B).
(PDF)

**S9 Fig. Correlations between estimated parameters.** The color and magnitude of each point shows the strength of the correlation; those with $p$-values greater than a significance threshold of 0.05 are crossed out. $p$-values were adjusted using the Benjamini-Hochberg correction. The strong correlation between $\beta_0$ and $d$ was included in the nonlinear mixed effects model framework.
(PDF)

**S10 Fig. Longer duration of maternal ART, and poor adherence, are associated with greater VL production to decay ratios.** The ratio is given by $\bar{p} = p/c$, in copies ml$^{-1}$ cell$^{-1}$. Each point represents a different infant.
(PDF)

**S11 Fig. The frequency of infection in CD4 T cells is usually small.** Distribution of the maximum proportion of total CD4 T cells that are infected ($I(t)/(I(t) + T(t))$) across all infants.
(PDF)

## Acknowledgments

Latency and Early neOnatal Provision of Anti-Retroviral Drugs (LEOPARD) Study Team:
Louise Kuhn, Elaine Abrams, Wei-Yann Tsai, Stephanie Shiau, Caroline Tiemessen, Maria Paximadis, Sharon Shalekoff, Diana Schramm, Gayle Sherman, Renate Strehlau, Megan Burke, Martie Conradie, Ashraf Coovadia, Ndileka Mbete, Faeezah Patel, Karl Technau, Grace Aldrovandi, Rohan Hazra, Devasena Gnanashanmugam.
The EPIICAL Consortium study team:
Nigel Klein, Diana Gibb, Sarah Watters, Man Chan, Laura McCoy, Abdel Babiker, Anne-Genevieve Marcelin, Vincent Calvez, Maria Angeles Munoz, Britta Wahren, Caroline Foster, Mark Cotton, Merlin Robb, Jintanat Ananworanich, Polly Claiden, Deenan Pillay, Deborah Persaud, Rob J de Boer, Juliane Schröter, Anet J N Anelone, Thanyawee Puthanakit, Adriana Ceci, Viviana Giannuzzi, Kathrine Luzuriaga, Nicolas Chomont, Mark Cameron, Caterina Cancrini, Andrew J Yates, Louise Kuhn, Sinead E Morris, Avy Violari, Kennedy Otwombe,

Ilaria Pepponi, Francesca Rocchi, Stefano Rinaldi, Alfredo Tagarro, Maria Grazia Lain, Paula Vaz, Elisa Lopez, Tacita Nhampossa.

We are very grateful to Juliane Schröter and Rob de Boer for their collaboration and their critical reading and insights.

## Author Contributions

**Conceptualization:** Sinead E. Morris, Louise Kuhn, Andrew J. Yates.

**Data curation:** Renate Strehlau, Stephanie Shiau, Elaine J. Abrams, Caroline T. Tiemessen, Louise Kuhn.

**Formal analysis:** Sinead E. Morris.

**Funding acquisition:** Louise Kuhn, Andrew J. Yates.

**Investigation:** Sinead E. Morris.

**Methodology:** Sinead E. Morris, Renate Strehlau, Stephanie Shiau, Elaine J. Abrams, Caroline T. Tiemessen, Louise Kuhn, Andrew J. Yates.

**Project administration:** Andrew J. Yates.

**Software:** Sinead E. Morris.

**Supervision:** Andrew J. Yates.

**Visualization:** Sinead E. Morris.

**Writing – original draft:** Sinead E. Morris, Louise Kuhn, Andrew J. Yates.

**Writing – review & editing:** Sinead E. Morris, Renate Strehlau, Stephanie Shiau, Elaine J. Abrams, Caroline T. Tiemessen, Louise Kuhn, Andrew J. Yates.

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
