## [Decision Letter · Decision Letter 0]

29 Mar 2022

Dear Prof. Yates,

Thank you very much for submitting your manuscript "Healthy dynamics of CD4 T cells may drive HIV resurgence in perinatally-infected infants on antiretroviral therapy" for consideration at PLOS Pathogens. As with all papers reviewed by the journal, your manuscript was reviewed by members of the editorial board and by several independent reviewers. The reviewers appreciated the attention to an important topic. Based on the reviews, we are likely to accept this manuscript for publication, providing that you modify the manuscript according to the review recommendations.

All three authors agree this is an excellent and important paper, which develops a mathematical model that takes into account many parameters of viral dynamics, and fits it to unique pediatric HIV infection data. However, all three reviewers all made multiple (and often overlapping) comments on clarity and structure of the paper, which I am quite confident can be addressed to make this paper clearer. Reviewer 1 noted that the data that the authors rely on is not accessible - this is of course an important issue as well.

Sincerely,

Susan R. Ross, PhD

Section Editor

PLOS Pathogens

Susan Ross

Section Editor

PLOS Pathogens

Kasturi Haldar

Editor-in-Chief

PLOS Pathogens

orcid.org/0000-0001-5065-158X

Michael Malim

Editor-in-Chief

PLOS Pathogens

orcid.org/0000-0002-7699-2064

All three authors agree this is an excellent and important paper, which develops a mathematical model that takes into account many parameters of viral dynamics, and fits it to unique pediatric HIV infection data. However, all three reviewers all made multiple (and often overlapping) comments on clarity and structure of the paper, which I am quite confident can be addressed to make this paper clearer. Reviewer 1 noted that the data that the authors rely on is not accessible - this is of course an important issue as well.

Reviewer Comments (if any, and for reference):

Reviewer's Responses to Questions

**Part I - Summary**

Reviewer #1: This is an interesting well-done analysis of CD4 T cell dynamics in perinatally-infected infants on ART.

The authors develop a mathematical that incorporates drug adherence, the kinetics of CD4 T cell resurgence in infants on ART as well as previously described declines on CD4 T cell populations with age. They fit their new model to data from the LEOPARD study and find that their new model agrees well with the data, whereas earlier simpler models did not. The paper increases our understanding of pediatric HIV infection and the effects of early ART and introduces new equations to describe T cell dynamics in HIV-infected infants.

Despite being very enthusiastic about this work, I do have some technical comments that need to be addressed.

1. In your description of the model on lines 97 and 98 you say ��is the fraction of infected cells that become productively infected or die through abortive infection. However, if they die then they should not contribute to the productively infected cells, I. I think you should have the abortively infected cells in the (1-��� fraction and that you should refer to I as the productively infected cells

2. You correctly point out that protease inhibitor blocks the production of infectious virus. Thus, shouldn’t your V equation refer to infectious virus not total virus? You could then add an equation for non-infectious virus as in Perelson et al. Science 1996 and properly fit the measured RNA to the sum of infectious and non-infectious virus.

3. To be consistent with previous literature it would be helpful to use d for the death rate of productively infected cells rather than d.

4. The compound parameter c bar is really a scaled viral production rate as suggested by your description of it as the contribution of each infected cell to the total VL. As such, why not use the symbol p bar. To avoid any confusion with the parameter representing clearance.

5. Please explain how you decided to use a logit-normal distribution for a and normal distributions for r and TR based on your preliminary fits. Also, what was the pre-specified upper bound for a?

6. I was surprised that you r estimate of the death rate of infected cells was so much smaller than estimates in adults, 0.05 per day vs ~ 1 per day (cf Markowitz et al JVI 77:5037-5038 (2003). Are there other estimates of the death rate of infected cells in infants that are consistent with your estimate?

7. I was also surprised that your estimate of R0 at ART initiation was 0.35 given that the median age at ART initiation was 2.5 days. If infection occurred at or near the time of birth then I would expect that on average the virus to be growing exponentially and not decaying before ART. Are there studies showing viral dynamics in untreated infants? In SHIV infection of rhesus macaques viral load increase for the first 1-2 weeks after infection – Fig 1 in JCI Insight. 2021;6(23):e152526. Please discuss this issue.

8. It is not clear how one can get access to the data used in this analysis. A data availability statement is needed

Reviewer #2: This is a very well-written paper, fitting data from young HIV-infected children with an impressive and careful mathematical fitting procedure. The mathematical model is extremely well-chosen as it is simple enough to keep parameters identifiable and rich enough to be able to account for recovery of CD4+ T cells and the natural decline in CD4 T cell counts in young children. It is impressive how well this simple model captures the wide variation in trajectories of the viral load and the CD4 count in all these children.

Reviewer #3: In this paper, the authors analyze an unique data set of HIV treatment in infants with a mathematical model for the viral dynamics, which also includes the dynamics of the target cells (in this case, CD4+ T cells). The paper is well written and the modeling elegantly done. The main strengths include the data set used, the novel modeling approach, and the interesting results. The paper is clearly written and the scholarship is of high caliber. Still, the work could be improved with some clarifications and also reference to some previous key publications.

Overall, the study is important in the relatively less studied field of pediatrics HIV, with focus on viral dynamics, and could be important to better understand the responses to treatment in this subgroups of infected people. The study is also somewhat novel, due to the quality of the data set and the simultaneous consideration of viral loads and CD4 counts, and treatment adherence.

**Part II – Major Issues: Key Experiments Required for Acceptance**

Reviewer #1: not applicable

Reviewer #2: No major issues.

Reviewer #3: There are no extra experiments needed to complete the manuscript. However, I think that there are some important issues that would benefit from some clarification.

One of the main assumptions is implicit in the use of CD4 T-cell numbers per unit volume. Although this is common in models of viral dynamics, I wonder if the absolute changes occurring in infants are properly captured this way. For example, the CD4+ T-cell concentration (per unit volume) decreases in this age range, but the absolute number may be increasing. HIV infection may be more dependent on the total available target cells and not their concentration. At least some thoughtful discussion of this issue would be welcome.

Related, in part, to the previous comment, there is important work in this area that should be referenced in the current paper. See for example the papers by Lewis, Callard et al in Frontiers Immuno 2017 and in J Infect Dis 2012, among others for their treatment of CD4+ T-cell levels; and the paper by Nagaraja, Gopalan, et al Scientific Reports 2021; or the older Melvin, Rodrigo et al JAIDS 1999.

One of the most difficult things in a long-term study of this kind in the change in therapy protocols that occur either by medical prescription or adherence issues. It would be useful to have some discussion of what the impact of using the average adherence for each infant is, as opposed to using their time series of adherence, which might not be possible (the model would be too complex for the data). However, I must say that it was not clear at all how you actually used the adherence (even the “average adherence”) of each infant in the model – do epsilon1 and/or epsilon2 (or their surrogates beta0 or c-bar) vary in time or are they estimated differently for different children? Was this used as a covariate (although it is not indicated in Table S1)? I couldn’t find how you used the adherence information. The available information itself is nicely shown in figures S9 and S10, and described in the first section of the supplementary material. But how was it used in the modeling?

It is not clear what is the impact and the need to consider the latent reservoir, namely the inclusion in the model of rho and a. Can you fit the data when rho= 1 and/or when a= 0?

Related with the previous comment, you state in line 199 that “in almost all infants the importance of new infection events was eventually superseded by reactivation from the latent reservoir”. However, this is only true when there is no more measurable virus (in the next sentence it even says that “this displacement was delayed by viral resurgence events…”). So, how relevant is the infection from the latent reservoir in the context of your study. Are these needed only for virus rebound when treatment is stopped?

One of your findings is that longer duration of maternal prenatal ART was associated with larger production of virus and with viral resurgence. This seems counter intuitive, and a little more discussion would be helpful. Is it that longer treatment may lead to transmission of resistant strains? In the discussion (line 286), you do say that longer prenatal ART is associated with larger viral reservoir in the infants. This is perhaps more puzzling that explanatory.

In your final conclusion, you state that resurgence may be inevitable in certain parameter regimes. This was not clear. What about increasing the adherence or improving efficacy of the drug protocol? The latter is a composed parameter, so perhaps you can't check in your model. But you could simulate the first effect for a couple of infants with resurgence to check what would happen assuming perfect adherence and keeping other parameters the same.

**Part III – Minor Issues: Editorial and Data Presentation Modifications**

Reviewer #1: No editorial or data presentation issue

Reviewer #2: Line 104 Model: The cells undergoing abortive infection are included into the population of infected cells (that in this paper have a long expected life span). An alternative way of writing this would be to include the abortive infections in the rho parameter, i.e., cells undergoing abortive infection fail to arrive in the compartment of productively infected cells (which has been done previously). Would that make any difference, and what is the most natural way to write this?

Line 151: Now rho=0.9999. If this were to include abortive infections it would be much smaller.

Line 208: What is the R0 in the absence of ART according to these parameters? Is that a reasonable value?

Line 218 and Fig 4B: Is there some circularity in this reasoning: children experiencing a resurgence in the VL necessarily have a slower average downslope of the VL, i.e., a lower production to decay ratio?

Lines 267-268: Is drug adherence not correlated for these 3 compounds?

Fig S5c: I agree this correlation is due to the outliers. Are these parametric correlations? A non-parametric test could be more careful.

Fig S9: Difficult to get a message from this, is Fig S10 not sufficient?

Reviewer #3: Line 60: interquartile range is formally a single number, the range, although I do recognize that it is often used in the way presented here, i.e., the 25th and 75th percentiles.

Line 111: Using c-bar is not a great choice, because it gives the impression that this parameter is related to c (the clearance rate), but in fact it is inversely proportional to c, and you interpret it as the “contribution of each infected cell to the total viral load”, so more like p (production rate). It seems that p-bar would be easier for the reader.

Line 145: please add the word “structural” to your description of “We confirmed the identifiability of all parameters…”.

Table 1 (and also in some parts of the text, e.g., line 206): you present the “SE= standard error (random effect)”. But Monolix usually (by default) presents the SE of the fixed effect, and the standard deviation of the random effect. Please confirm which one of these you are discussing, or if indeed it is the standard error of the random effects, which is something different and indeed probably not very appropriate for this situation.

Line 153: You used the general AIC formula. Usually, it is recommended to use the small sample corrected AIC, even when the sample seems large enough, because you are also estimating many parameters (in Table 1, 19 parameters or so, including error and correlation parameters). However, technically the small sample AIC formula does not apply directly to multivariate models (and fits), as is the case here and further corrections are needed. See for example the book by Burnham and Anderson (section 7.7.6). This is just something for you to consider what is the best option in this case.

Figures with time courses: I suggest that you plot the figure with the line on top of the data, rather than the data on top of the line. The latter makes it difficult to see the fits when there is a concentration of data points.

Line 219: what do you mean with “deterministic model parameters”, deterministic as opposed to what?

Figure 4: Please define the labels in the x axis: "Before" - before being pregnant? Define "12+ wks" more than twelve weeks of treatment or initiated at more than 12 weeks of pregnancy? This only become clear when consulting table S1.

Figure 4 caption: description of panel (F), the words “is associated” seem extra.

Line 310: You state that “frequency of infection in CD4 T cells is usually small”, but was this true in your case. That is, what is the time course of I(t)/(T(t)+I(t)) in your results?

Table S1: It is not clear that “Variable treatment: continuous” means when you describe the variable as grouped. For example, “Birth weight” was divided into two groups more or less than 2500g. How do you treat this as a continuous covariate in Monolix? The same question for “Mother’s viral load”, etc…

Figure S7: The symbols seem to have different sizes, but this is not described in the caption.

PLOS authors have the option to publish the peer review history of their article (what does this mean?). If published, this will include your full peer review and any attached files.

Reviewer #1: No

Reviewer #2: **Yes: **Rob de Boer

Reviewer #3: No

Figure Files:

Data Requirements:

Reproducibility:

References:

---

## [Decision Letter · Decision Letter 1]

14 Jul 2022

Dear Prof. Yates,

Thank you very much for submitting your manuscript "Healthy dynamics of CD4 T cells may drive HIV resurgence in perinatally-infected infants on antiretroviral therapy" for consideration at PLOS Pathogens. As with all papers reviewed by the journal, your manuscript was reviewed by members of the editorial board and by several independent reviewers. The reviewers appreciated the attention to an important topic. Based on the reviews, we are likely to accept this manuscript for publication, providing that you modify the manuscript according to the review recommendations.

In addition to the minor revisions requested by reviewer 3, please make sure that there is a statement in the manuscript regarding the public availability of your data.

Sincerely,

Susan R. Ross, PhD

Section Editor

PLOS Pathogens

Susan Ross

Section Editor

PLOS Pathogens

Kasturi Haldar

Editor-in-Chief

PLOS Pathogens

orcid.org/0000-0001-5065-158X

Michael Malim

Editor-in-Chief

PLOS Pathogens

orcid.org/0000-0002-7699-2064

Reviewer Comments (if any, and for reference):

Reviewer's Responses to Questions

**Part I - Summary**

Reviewer #1: This is a thorough analysis of the dynamics of CD4 T cells in perinatally infected infants on ART. The approach used is novel and should be of general interest.

Reviewer #2: See my previous review: this is an excellent paper.

Reviewer #3: (No Response)

**Part II – Major Issues: Key Experiments Required for Acceptance**

Reviewer #1: No major issues or new experiments needed.

Reviewer #2: See my previous review: this is an excellent paper.

Reviewer #3: (No Response)

**Part III – Minor Issues: Editorial and Data Presentation Modifications**

Reviewer #1: No issues

Reviewer #2: All my recommendations have been handled carefully.

Reviewer #3: I thank the authors for the careful revision of the manuscript. I have only a couple of very minor questions.

1) Page 9, line 231: I believe this should be "standard error of the fixed effects" and not random effects. Please re-check throughout.

2) Page 9, line 234: are these the limits of the interval, or the range of multiple intervals across the children.

3) Page 9, line 258: instead of p, it would me more appropriate to say "not significant at the alpha=0.05 level."

4) Page 15, line 348: regarding your estimate of 16 days, there is a new paper that may be relevant, see White et al. PNAS 2022: https://doi.org/10.1073/pnas.2120326119

PLOS authors have the option to publish the peer review history of their article (what does this mean?). If published, this will include your full peer review and any attached files.

Reviewer #1: No

Reviewer #2: **Yes: **Rob J de Boer

Reviewer #3: No

Figure Files:

Data Requirements:

Reproducibility:

References:

---

## [Editor Report · Decision Letter 2]

19 Jul 2022

Dear Prof. Yates,

We are pleased to inform you that your manuscript 'Healthy dynamics of CD4 T cells may drive HIV resurgence in perinatally-infected infants on antiretroviral therapy' has been provisionally accepted for publication in PLOS Pathogens.

Best regards,

Susan R. Ross, PhD

Section Editor

PLOS Pathogens

Susan Ross

Section Editor

PLOS Pathogens

Kasturi Haldar

Editor-in-Chief

PLOS Pathogens

orcid.org/0000-0001-5065-158X

Michael Malim

Editor-in-Chief

PLOS Pathogens

orcid.org/0000-0002-7699-2064
---

## [Editor Report · Acceptance letter]

9 Aug 2022

Dear Prof. Yates,

We are delighted to inform you that your manuscript, "Healthy dynamics of CD4 T cells may drive HIV resurgence in perinatally-infected infants on antiretroviral therapy," has been formally accepted for publication in PLOS Pathogens.

Best regards,

Kasturi Haldar

Editor-in-Chief

PLOS Pathogens

orcid.org/0000-0001-5065-158X

Michael Malim

Editor-in-Chief

PLOS Pathogens

orcid.org/0000-0002-7699-2064